# Comparative Roles of Rad4A and Rad4B in Photoprotection of *Beauveria bassiana* from Solar Ultraviolet Damage

**DOI:** 10.3390/jof9020154

**Published:** 2023-01-23

**Authors:** Lei Yu, Si-Yuan Xu, Xin-Cheng Luo, Sheng-Hua Ying, Ming-Guang Feng

**Affiliations:** Institute of Microbiology, College of Life Sciences, Zhejiang University, Hangzhou 310058, China

**Keywords:** filamentous fungi, anti-UV proteins, photoreactivation, nucleotide excision repair

## Abstract

The Rad4-Rad23-Rad33 complex plays an essential anti-ultraviolet (UV) role depending on nucleotide excision repair (NER) in budding yeast but has been rarely studied in filamentous fungi, which possess two Rad4 paralogs (Rad4A/B) and orthologous Rad23 and rely on the photorepair of UV-induced DNA lesions, a distinct mechanism behind the photoreactivation of UV-impaired cells. Previously, nucleocytoplasmic shuttling Rad23 proved to be highly efficient in the photoreactivation of conidia inactivated by UVB, a major component of solar UV, due to its interaction with Phr2 in *Beauveria bassiana*, a wide-spectrum insect mycopathogen lacking Rad33. Here, either Rad4A or Rad4B was proven to localize exclusively in the nucleus and interact with Rad23, which was previously shown to interact with the white collar protein WC2 as a regulator of two photorepair-required photolyases (Phr1 and Phr2) in *B. bassiana*. The Δ*rad4A* mutant lost ~80% of conidial UVB resistance and ~50% of activity in the photoreactivation of UVB-inactivated conidia by 5 h light exposure. Intriguingly, the reactivation rates of UVB-impaired conidia were observable only in the presence of *rad4A* after dark incubation exceeding 24 h, implicating extant, but infeasible, NER activity for Rad4A in the field where night (dark) time is too short. Aside from its strong anti-UVB role, Rad4A played no other role in *B. bassiana*’s lifecycle while Rad4B proved to be functionally redundant. Our findings uncover that the anti-UVB role of Rad4A depends on the photoreactivation activity ascribed to its interaction with Rad23 linked to WC2 and Phr2 and expands a molecular basis underlying filamentous fungal adaptation to solar UV irradiation on the Earth’s surface.

## 1. Introduction

An all-weather application of environmentally friendly fungal formulations for arthropod pest control is desired but restrained in hot summer, which features strong sunlight and high temperature. Upon field application, formulated conidia are inevitably exposed to solar ultraviolet (UV) irradiation composed of UVB (280–320 nm) and UVA (320–400 nm) wavelengths [1], which cause cellular damages by impairment of intracellular macromolecules and generation of reactive oxygen species, respectively [2,3]. Therefore, it is necessary to understand the molecular mechanisms behind fungal adaptation to solar UV that are detrimental to fungal conidia [4,5,6]. The understanding will help to improve field application strategies of fungal pesticides in summer [7,8].

Damages of eukaryotic cells exposed to UV irradiation arise mainly from the generation of UV-induced DNA lesions known as cytotoxic cyclobutane pyrimidine dimer (CPD) and (6-4)-pyrimidine-pyrimidone (6-4PP) photoproducts [2,3,9,10,11]. Such cytotoxic photoproducts form through the covalent linkages of adjacent bases in DNA duplexes under the irradiation. As far as has been learned to date, fungi have evolved two distinct mechanisms, namely rapid photorepair and slow nucleotide excision repair (NER), to recover the CPD and 6-4PP DNA lesions [3,12]. Photorepair is a process of repairing shorter UV-induced DNA lesions by direct transfer of electrons to the CPD or 6-4PP lesions to cut off the covalent linkages under longer UV or visible light [13,14]. Filamentous fungal photorepair proceeds under the action(s) of one or two photolyases, i.e., CPD-specific Phr1 and/or 6-4PP-specific Phr2, instead of one or two cryptochromes called Cry-DASHs [15,16,17,18,19]. Indeed, both photolyases and Cry-DASHs are classified to the same family consisting of no more than four members, which share a DNA_Photolyase domain required for photorepair activity. The dependence of filamentous fungal photorepair activity on photolyase(s) instead of Cry-DASH(s) has been long unclear. This puzzle has been solved in a recent study showing the dependence of photorepair on nuclear localization of either Phr1 or Phr2 rather than on cytoplasmic localization of unique Cry-DASH (CryD) in *Beauveria bassiana* [19]. The occurrence of photorepair as a nucleus-specific event is evidenced with a nuclear localization signal (NLS) motif that is shared by all photolyase homologs at high probabilities but is not predictable in any true Cry-DASH sequences of many fungal genomes surveyed [7].

Unlike rapid photorepair, NER proceeds slowly in full darkness and depends on the activities of many more enzymes and proteins that are involved in poly-ubiquitination and proteasome activity and have been intensively studied in *Saccharomyces cerevisiae* [12,20,21,22,23]. In the yeast, many anti-UV radiation (RAD) genes identified in early studies [24,25] code for a large family of RAD proteins that function in the NER pathway to mediate the global-genome NER (GG-NER) of UV-induced DNA lesions [26]. Indeed, multiple RAD protein complexes play essential roles in the yeast GG-NER process. For instance, Rad1 and Rad10 interact with each other to form the Rad1-Rad10 complex serving as an endonuclease that can recognize the junction between single- and double-strand DNAs and remove unpaired 3′ tails from the junction [27,28,29]. Interactions of Rad4 with Rad23 and Rad33 result in the formation of a Rad4-Rad23-Rad33 complex enabling the sensing of a distorted DNA duplex for commencement of GG-NER [30] by its recruitment to impaired DNA sites through interactions of Rad4 with chromatin remodeling complexes [21,31]. However, NER has been rarely studied in fungi other than the model yeast, making it elusive as to whether NER is functional in filamentous fungal pathogens of plants and insects on the Earth’s surface. Notably, previous studies on either yeast NER or filamentous fungal photorepair were based on an artificial irradiation source of UVC (<280 nm), which does not exist on the Earth’s surface due to complete removal from solar irradiation by atmospheric ozone [1] but is far more harmful to fungal cells than UVB, a major component of solar UV. It is necessary to simulate a field situation, where UVB is a main source of fungal DNA lesions, in an attempt to explore molecular mechanisms underlying the reactivation of formulated conidia impaired by solar UV.

Filamentous fungi possess homologs of most anti-UV RAD proteins that function in the yeast NER pathway [26]. However, none of them had been studied until recently. For the Rad23 required for the yeast GG-NER [21,32,33], its ortholog interacting with Phr2 was able to reactivate most of the UVB-inactivated conidia after 3 h exposure to visible light but was unable to do so after 24 h dark incubation in *B. bassiana* [34]. The light- reactivating process of conidia impaired or inactivated by UV is known as photoreactivation, which is an output, and hence an index, of photorepair activity. WC1 and WC2, two white collar proteins lacking a DNA_Photolyase domain, were shown to function as well as or better than Phr1/2 in the photorepair of UVB-induced CPD and 6-4PP DNA lesions because both of them acted as regulators of Phr1/2 in *B. bassiana* and *Metarhizium robertsii* [35,36], two hypocrealean insect pathogens serving as the main sources of fungal pesticides worldwide [37]. For Rad1 and Rad10 essential for the yeast GG-NER, their orthologs were found to interact with WC2 or Phr1 in *M. robertsii* and also with both WC1 and WC2 in *B. bassiana*, and hence to have acquired high photoreactivation activities in both insect pathogens [36,38]. Interestingly, both the Rad1 and Rad10 orthologs showed extant NER activity when dark incubation was prolonged to 40 h [36]. Nonetheless, the NER activity was incapable of reactivating the UVB-impaired conidia of either insect pathogen after less than 24 h dark incubation, suggesting that an infeasibility of filamentous fungal NER is ascribed to too short dark (night) time available for NER in the field. In filamentous fungi, WC1 and WC2 interact with each other to form a white collar complex (WCC), which is a well-known regulator of manifold genes involved in response to light [39] and plays an essential role in running a circadian clock [40,41,42]. The recent progresses in fungal insect pathogens largely expanded a molecular basis of filamentous fungal photorepair that was long considered to depend on only one or two photolyases and provided strong evidence for a hypothesis that photorepair could be the sole feasible mechanism behind filamentous fungal adaptation to solar UV and is mechanistically far more complicated than what was learned prior to the recent studies [7]. In other words, WCC could serve as a core of the feasible mechanism to mediate the expression levels of various anti-UV genes encoding not only photolyases but also RAD proteins and associated partners. Following an elucidation of the Rad23 ortholog’s role in photoreactivation and of its pleiotropic effect in the fungal lifecycle [34], this study sought to further test the hypothesis by functional characterization of two Rad4 paralogs (Rad4A and Rad4B) in *B. bassiana* lacking a protein homologous to the yeast Rad33. The coexistence of two Rad4 paralogs and the Rad23 ortholog and the absence of the Rad33 homolog in *B. bassiana* hint at a scenario distinct from the Rad4-Rad23-Rad33 complex elucidated in the model yeast [21,30,31]. Our data reveal an interaction between either Rad4A or Rad4B and Rad23 and a high activity of only Rad4A in photoreactivation rather than in NER-depending reactivation but functional redundancy of Rad4B in *B. bassiana*, providing more evidence for the hypothesis.

## 2. Materials and Methods

### 2.1. Bioinformatic Analysis of Fungal Rad4 Homologs

BLASTp analysis (http://blast.ncbi.nlm.nih. gov/blast.cgi, accessed on 23 January 2023) was carried out to search through the NCBI databases of selected ascomycetes using *S. cerevisiae* Rad4 (NP_011089) as a query. Conserved domains (http://smart.embl- heidelberg.de/, accessed on 23 January 2023) and NLS motifs with maximal probability (https://www.novopro.cn/tools/nls-signal-prediction, accessed on 23 January 2023) were predicted from the located homologs of *B. bassiana* and the yeast query and also from the yeast Rad23 (NP_010877) and its Rad23 ortholog (EJP70161) characterized previously in *B. bassiana* [34], respectively, for a comparison with each other. All Rad4 homologs found in the ascomycetes were clustered using the maximum likelihood method in the online program MEGA11 (http://www.megasoftware.net/, accessed on 23 January 2023).

### 2.2. Subcellular Localization of Rad4A, Rad4B and Rad23 in B. bassiana

Due to an essentiality of nuclear localization for DNA-repair cellular events, the subcellular localization of Rad4A and Rad4B was explored using the green fluorescence protein GFP-tagged fusion protein of each expressed in the wild-type strain *B. bassiana* ARSEF 2860 (denoted as WT hereafter) as described previously [36]. For each target gene, the open reading frame (ORF) amplified from the WT cDNA with paired primers (Appendix A) was ligated to the 5′-terminus of *gfp* (U55763) in the linearized pAN52-gfp-bar vector using a one-step cloning kit (Vazyme, Nanjin, China). The constructed vector pAN52-*x*- gfp-bar (*x* = *rad4A* or *rad4B*) controlled by the endogenous promoter *Ptef1* was integrated into the WT genome via a transformation mediated by *Agrobacterium*. Transgenic colonies were screened via the *bar* resistance to phosphinothricin (200 μg/mL) and examined under a fluorescence microscope. Based on the desired green signal, a colony selected from those generated by each transformation was incubated for conidiation on SDAY (Sabouraud dextrose agar (4% glucose, 1% peptone and 1.5% agar) plus 1% yeast extract) at the optimal regime of 25 °C and 12 h light:12 h dark. The resultant conidia were suspended in SDBY (agar-free SDAY), followed by a 3-day incubation on a shaking bed (150 rpm) at 25 °C. Hyphal samples taken from the cultures were stained with 4.16 mM DAPI (4′,6′-diamidine-2′-phenylindole dihydrochloride; Sigma-Aldrich, Shanghai, China) and visualized for subcellular localization of Rad4A-GFP or Rad4B-GFP via laser scanning confocal microscopy (LSCM) at the excitation/emission wavelengths of 358/460 and 488/507 nm. For either fusion protein, green fluorescence intensity was assessed from a fixed circular area moving in the cytoplasm and nucleus of each of 15 hyphal cells using ImageJ software (https://imagej.nih.gov/ij/, accessed on 23 January 2023) and used to compute nuclear versus cytoplasmic green fluorescence intensity (N/C-GFI) ratio as its relative accumulation level in the nucleus of each hyphal cell.

Co-localization of either Rad4A or Rad4B with Rad23 was explored to reveal a possible interaction of either with Rad23. Briefly, the *rad23* ORF was ligated to the 5′-terminus of *mCherry* (KC294599) in the vector pAN52-mCherry-sur under the control of *Ptef1* [36]. The new vector was transformed into the respective strains expressing Rad4A-GFP and Rad4B-GFP as aforementioned, followed by the screening of transgenic colonies via *sur* resistance to chlorimuron ethyl (10 μg/mL). A colony showing desirable red fluorescence was chosen for subcellular co-localization of Rad4A-GFP or Rad4B-GFP with Rad23-mCherry by LSCM at the excitation/emission wavelengths of 488/507 and 561/610 nm.

### 2.3. Yeast Two-Hybrid (Y2H) Assays

The protein–protein interactions among Rad4A, Rad4B, Rad23, Phr1, Phr2, WC1 and WC2 were detected using our previous Y2H protocol [35,36]. Briefly, the ORFs of *rad4A* (BBA_02814), *rad4B* (BBA_02963), *rad23* (BBA_01030), *phr1* (BBA_01664), *phr2* (BBA_01034), *wc1* (BBA_10271) and *wc2* (BBA_01403) were amplified from the WT cDNA and ligated to the prey vector pGADT7 (AD) or the bait vector pGBKT7 (BD). The constructed vectors were verified via sequencing, transformed into the *S. cerevisiae* Y187 and Y2HGold strains, respectively, and incubated at 30 °C for 24 h with pairwise yeast mating on YPD (1% yeast extract, 2% peptone, 2% glucose plus 0.04% adenine hemisulfate salt). A positive control (AD-LargeT-BD-P53) and multiple negative controls (empty AD-BD and semi-empty AD-*x*-BD or AD-BD-*x* constructs, where *x* denotes one of the target proteins) were included in the screening of the yeast diploids on the plates of a synthetically defined medium (SDM) with nutritional double- (SDM/-Leu/-Trp/X-α-Gal/AbA) and quadruple-dropout (SDM/-Leu/-Trp/-Ade/-His/X-α-Gal/AbA). All yeast colonies initiated by spotting 5 × 10^2^, 5 × 10^3^ and 5 × 10^4^ cells on the double- and quadruple-dropout plates were incubated for 3 days at 30 °C.

### 2.4. Generation of rad14A and rad4B Mutants

The gene *rad4A* or *rad4B* was disrupted in the WT strain by deleting a DNA fragment comprising a small part of the promoter region and most of the *rad4A* (2281 bp) or *rad4B* (2513 bp) coding regions (Appendix A) via homologous recombination of the *bar*-separated 5′ and 3′ coding/flanking fragments of the constructed vector p0380-5′*x*- bar-3′*x* (*x* = *rad4A* or *rad4B*). Either target gene was complemented into its identified null mutant by integrating ectopically a cassette comprising flanking and a full-coding sequence of *rad4A* (6107 bp) or *rad4B* (6100 bp) and a *sur* marker in the vector p0380-*x*-sur. The constructed vectors verified via sequencing were transformed into the corresponding WT or null mutant, and putative mutant colonies were screened via the *bar* resistance and the *sur* resistance, respectively, as aforementioned. The expected recombination events were analyzed via PCR (Appendix A) and verified via real-time quantitative PCR (qPCR) analysis (Appendix A). All paired primers used for the manipulation and detection of each target gene are listed in Appendix A. The disruption mutants (DMs) and the complementation mutants (CMs) of *rad4A* and *rad4B* and the WT strain were used in the following experiments with each including three independent replicates.

### 2.5. Assays for Fungal Lifecycle-Related Phenotypes

All lifecycle-related phenotypes were examined as described previously [36]. The radial growth of each fungal strain was initiated with 10^3^ conidia on rich SDAY, 1/4 SDAY (amended with one quarter strength of each SDAY nutrient), minimal Czapek-Dox agar (CDA: 3% sucrose, 0.3% NaNO_3_, 0.1% K_2_HPO_4_, 0.05% KCl, 0.05% MgSO_4_ and 0.001% FeSO_4_ plus 1.5% agar) and CDAs amended with different carbon or nitrogen sources, or by removing the carbon or nitrogen source. Stress assays were performed by initiating the radial growth of each strain on the plates of CDA supplemented with different types of chemical stressors or by exposing 2-day-old SDAY colonies to 42 °C heat shock for 3–9 h. In addition, SDAY and CDA plates inoculated with 10^3^ conidia for the initiation of colony growth were exposed to a UVB irradiation of 0.2 J/cm^2^ (detailed later). After 7-day incubation or 5-day growth recovery (after the heat shock) at the optimal regime, the diameter of each colony was measured as a growth index under normal conditions or different stresses. The relative growth inhibition [RGI = (*d*_c_ − *d*_s_)/*d*_c_ × 100)] of each strain was estimated as an index of its sensitivity to a given stress using the diameters of stressed (*d*_s_) and control (*d*_c_) colonies. Conidial yields were assessed as the number of conidia per square centimeter from three 7-day-old SDAY cultures of each strain initiated by spreading 100 μL of a 10^7^ conidia/mLsuspension per plate (*ϕ* = 9 cm) at the optimal regime.

The virulence of each strain against the model insect *Galleria mellonella* (fourth-instar larvae) was assayed by immersing three groups of ~35 larvae for 10 s in 40 mL aliquots of a 10^7^ conidia/mL suspension for normal cuticle infection or by intrahemocoel injection of ~500 conidia (in 5 μL of a 10^5^ conidia/mL suspension) into the hemocoel of each larva in each group for cuticle-bypassing infection. After inoculation, all groups were held at 25 °C for survival/mortality records every 12 or 24 h. An estimate of median lethal time (LT_50_, no. days) was made as a virulence index of each strain against the model insect via modeling analysis of the resultant time-mortality trend in each group of larvae infected in either mode.

### 2.6. Assays for Conidial UVB Resistance

Our previous protocol [43] was adopted to assay conidial UVB resistance of each strain in a Bio-Sun^++^ UV irradiation chamber (Vilber Lourmat, Marne-la-Vallée, France), in which an inset microprocessor automatically adjusts the irradiating wavelength (weighted 312 nm) and intensity four times per second to control an error of ≤1 μJ/cm^2^ (10^−6^) for a given dose of irradiation (according to the manufacturer’s guide). Briefly, three 100 μL aliquots of a 10^7^ conidia/mL suspension per strain were spread onto the plates of a germination medium (GM; 2% sucrose and 0.5% peptone plus 1.5% agar) and exposed to UVB irradiation at the gradient doses of 0.02, 0.03, 0.07, 0.1, 0.15, 0.2, 0.25, 0.3, 0.35 and 0.4 J/cm^2^, respectively, in the sample tray of the chamber after air drying of sterile ventilation for 10 min to minimize the impact of moisture on the irradiation. The irradiated plates were covered immediately with lids and incubated at 25 °C for 24 h in the dark. Three plates not irradiated were used as a control. From 8 h dark incubation onwards, three microscopic view fields (100× magnification) per plate were observed every 2 h to monitor a maximal germination rate of the conidia irradiated at each UVB dose. The ratio of maximal germination percentages of irradiated versus non-irradiated conidia was computed as the conidial survival index (*I*_s_) with respect to the control. The observed *I*_s_ trend over the gradient doses fitted a modified logistic equation [43]. The fitted equations of three replicates per strain were used to estimate the indices of conidial UVB resistance as lethal doses to inactivate 50% (LD_50_), 75% (LD_75_) and 95% (LD_95_) of conidia at the end of 12 and 24 h dark incubation after UVB irradiation.

### 2.7. Assays for Photoreactivation and NER Activities

Assays for photoreactivation and NER activities of fungal strains were conducted as described previously [19,36]. Briefly, GM plates smeared evenly with100 μL aliquots of the conidial suspension were irradiated at the respective UVB doses of 0.2, 0.3 and 0.4 J/cm^2^ in the UV chamber. The irradiated plates were incubated at 25 °C for 5 h under white light and then 19 h in the dark (photoreactivation treatment) or directly for up to 40 h in the dark (NER treatment). Maximal germination percentages of the irradiated conidia were monitored as aforementioned during the period of dark incubation following the first 5 h incubation under light or in the dark.

### 2.8. qPCR Analysis

For all DM and control (WT and CM) strains, 100 μL aliquots of a 10^7^ conidia/mL suspension were spread onto cellophane-overlaid SDAY plates and incubated for up to 7 days at the optimal regime. Total RNAs were extracted daily from the 2- to 7-day-old WT cultures and from the 3-day-old cultures of each DM or control strain using an RNAiso Plus Kit (TaKaRa, Dalian, China) and reversely transcribed into cDNAs using a PrimeScript RT reagent kit (TaKaRa). Transcripts of target genes in the cDNA samples were quantified using qPCR analysis with paired primers (Appendix A) under the action of SYBR Premix *ExTaq* (TaKaRa), including the fungal 18S rRNA used as a reference. A threshold cycle (2^−ΔΔCt^) method was used to assess (1) relative transcript levels of *rad4A*, *rad4B* and *rad23* in the WT cultures over the days of incubation with respect to a standard at the end of 2-day cultivation; (2) relative transcript levels of *rad4A* and *rad4B* in the 3-day-old cultures of their DM and CM strains with respect to the WT standard in order to verify expected recombination events in the mutants; and (3) relative transcript levels of several photorepair- or photoreactivation-related genes in the 3-day-old cultures of the DM and CM strains with respect to the WT standard. A onefold transcript change was used as a standard of significant down- or upregulation.

### 2.9. Statistical Analysis

Student’s *t* test was carried out to reveal a difference between the N/C-GFI ratios of Rad4A-GFP and Rad4B-GFP expressed in the WT strain. A one-way analysis of variance was performed to differentiate between a variation in data from each phenotypic experiment comprising three independent replicates and mean parameters of all tested strains on the basis of Tukey’s honestly significant difference (HSD).

## 3. Results

### 3.1. Recognition and Domain Architecture of Fungal Rad4 Paralogs

The BLASTp analysis using *S. cerevisiae* Rad4 as a query resulted in the location of two Rad4 paralogs (Rad4A: EJP67918; Rad4B: EJP68067) in *B. bassiana* and other ascomycetes. The *B. bassiana* Rad4A (839 amino acids (aa)) and Rad4B (907 aa) share sequence identities of 31.54% (with a total score of 276, a coverage of 74% and an e-value of 3 × 10^−53^) and 27.64% (with a total score of 158, a coverage of 74% and an e-value of 1 × 10^−28^) with the yeast Rad4, respectively. Through conserved domain analysis, Rad4A and Rad4B were revealed to share a typical Rad4 domain and three Rad4 beta-hairpin domains (BH_D1, BH_D2 and BH_D3) with the yeast query but lack an additional Transglut_core domain present in the yeast query (Figure 1A). An NLS motif was predicted from the amino acid sequence of Rad4A, Rad4B and the yeast query at the high probabilities of 0.715–0.985. In contrast, the NSL motif was predicted at the low probability of only 0.129 from the sequence of the Rad23 ortholog studied previously in *B. bassiana* [34] but not predictable from the yeast Rad23 sequence. Interestingly, the use of the yeast query in the BLASTp analysis uncovered a wide existence of two Rad4 paralogs in the fission yeast- inclusive ascomycetes except for *Candida albicans*, which shares a Rad4 ortholog with *S. cerevisiae*. In phylogeny, both Rad4A and Rad4B homologs are closely related to the lineages of fungal species (Appendix A). These analyses reveal that two Rad4 paralogs exist widely in filamentous fungi and that Rad4A and Rad4B in *B. bassiana* are similar to the yeast query in domain architecture despite a difference between their molecular sizes.

### 3.2. Expression and Localization of Rad4A and Rad4B in B. bassiana

The coding genes of Rad4A and Rad4B were constitutively expressed like *rad23* in WT cultures during the period of 7-day incubation on SDAY at the optimal regime of 25 °C and 12:12 (L:D) (Figure 1B). Compared to the standard level on Day 2, *rad4A* and *rad23* were increasingly upregulated during the first 5-day incubation but then expressed differentially. The expression level of *rad4B* was not upregulated until Day 4 and was consistently lower than that of *rad4A* during the period of incubation. Overall, *rad4A* was more active than *rad4B* in the WT strain at the transcriptional level.

The LSCM images showed a nucleus-specific localization of either the Rad4A-GFP or Rad4B-GFP fusion protein expressed in the WT hyphae, as indicated by an orange color well merged from expressed green and DAPI-stained color (shown in red) in the nuclei of hyphal cells (Figure 1C). The mean (±SD) N/C-GFI ratios of 8.06 (±3.11) and 8.27 (±3.35) assessed from 15 hyphal cells expressing Rad4A-GFP and Rad4B-GFP, respectively, were similar to each other (*p* = 0.86 in Student’s *t* test) (Figure 1D). The images and the ratios confirm that both Rad4A and Rad4B are nucleus-specific proteins, coinciding well with the high probability of an NLS motif predicted for either. These observations suggest an involvement of either paralog in fungal nuclear events, such as DNA repair.

### 3.3. Co-Localization and Interaction of Rad4A or Rad4B with Rad23

An interaction between Rad4 and Rad23 is crucial to yeast NER activity [21,44]. In the present study, either Rad4A-GFP and Rad23-mCherry or Rad4B-GFP and Rad23-mCherry co-expressed in the WT strain were well co-localized in the nuclei of hyphal cells, although Rad23-mCherry was localized in both the nuclei and cytoplasm, as shown by LSCM images (Figure 1E,F). These observations implicated a possible interaction between Rad4A or Rad4B and Rad23 in *B. bassiana*.

The implication was verified by the following Y2H assays. The colonies of the yeast diploids grown on the quadruple-dropout plates confirmed an interaction either between Rad4A and Rad23 (Figure 2A) or between Rad4B and Rad23 (Figure 2B). A positive interaction was also detected between Rad23 and WC2 as a regulator of both *phr1* and *phr2* in *B. bassiana* [36], rather than between Rad23 and WC1, the other regulator of *phr1* and *phr2* (Figure 2B). However, no interaction was found between Rad4A and Rad4B (Appendix A), suggesting a functional independence of each other, nor were interactions detected between Rad4A or Rad4B and any photolyase (Phr1/2) or photolyase regulator (WC1/2) (Appendix A). Taking a previously detected Rad23-Phr2 interaction [34] into account, the protein–protein interactions detected suggest a bridge of Rad23 between a link of Rad4A or Rad4B to the photolyase and the photolyase regulator required for DNA photorepair in *B. bassiana* [19,36].

### 3.4. Dispensable Roles of Rad4A and Rad4B in Fungal Lifecycle

Almost all phenotypes associated with fungal lifecycle were not altered significantly in the DM strains of *rad4A* and *rad4B* (*p* > 0.05 in Tukey’s test) in comparison to their control strains (Appendix A). The examined phenotypes included radial growth on rich and minimal media under normal culture conditions (Appendix A), cellular tolerance to osmotic, oxidative, cell-wall-perturbing and heat-shock stresses (Appendix A), conidiation capacity (Appendix A) and virulence via the cuticle or cuticle-bypassing infection (Appendix A). As an exception, conidial germination was significantly slower in the absence of *rad4B*, although the change was minor (Appendix A). These observations demonstrated a dispensability of both Rad4A and Rad4B for the in vitro and in vivo lifecycle of *B. bassiana*.

### 3.5. Essential Role of Rad4A versus Null Role of Rad4B in Fungal Resistance to UVB

Despite a null response to the chemical stresses and heat shock, the *rad4A* DM strain’s radial growth on SDAY and CDA was abolished by an exposure to the UVB dose of 0.2 J/cm^2^ after inoculation with 10^3^ conidia (Figure 3A). In contrast, the DM strain of *rad4B* grew as well as the control strains on the plates exposed to the same UVB dose. This observation indicates an involvement of only Rad4A in cell resistance to UVB.

Next, conidial survival trends over the gradient UVB doses of 0.02–0.4 J/cm^2^ (Figure 3B) fitted well the logistic model at the respective ends of 12 and 24 h dark incubation (r^2^ ≥ 0.975 in fitness *F* tests). The mean estimates (*n* = 9) of LD_50_, LD_75_ and LD_95_ for the control stains’ conidia were 0.091 (±0.006), 0.128 (±0.008) and 0.191 (±0.011) J/cm^2^ at the end of 12 h dark incubation, and increased to 0.212 (±0.004), 0.254 (±0.006) and 0.325 (±0.012) J/cm^2^ at the end of 24 h dark incubation (Figure 3C), respectively. Compared to these estimates, the *rad4A* DM’s LD_50_, LD_75_ and LD_95_ were significantly reduced (*p* < 0.001 in Tukey’s test) by ~81% at the end of12 h dark incubation and by 83%, 78% and 73% at the end of 24 h dark incubation, respectively. In contrast, none of the estimates were affected in the absence of *rad4B*. The survival trends of the control strains’ conidia were differentially impaired as the gradient UVB doses and much lower at the end of the 12 h compared with the 24 h dark incubation, implicating the reliance of extant NER activity on the time length of dark incubation. The lethal doses that decreased greatly only in the absence of *rad4A* indicated a dependence of the extant NER activity on Rad4A rather than on Rad4B. Obviously, only Rad4A is essential for conidial UVB resistance in *B. bassiana*.

### 3.6. Rad4A Has High Photoreactivation Activity but Infeasible NER Activity

The dark condition is limited to daily nighttime of only 10 h or so in the summer on the Earth’s surface, suggesting an infeasibility for the NER activity of Rad4A against solar UV in the field. This inference was clarified by reactivating conidia differentially impaired at the UVB doses of 0.2, 0.3 and 0.4 J/cm^2^ under visible light and prolonged dark conditions. After UVB irradiation at 0.4 J/cm^2^, prompt 5 h light exposure plus 19 h dark incubation for photoreactivation at 25 °C resulted in the germination of UVB-inactivated conidia that were readily observable for all tested strains (Figure 4A). 

In contrast, reactivated (germinated) conidia of the control strains were hardly visible at the end of direct 24 h dark incubation for NER but became observable after longer dark incubation. Consequently, the control strains’ conidia impaired severely or inactivated at the UVB doses of 0.2, 0.3 and 0.4 J/cm^2^ were photoreactivated by 100%, 98.8% (±1.0) and 88.4% (±2.7) (*n* = 9), respectively (Figure 4B). The photoreactivation rates decreased significantly to 87.7% (±2.5), 59.7% (±2.5) and 43.3% (±3.1) in the *rad4A* DM (*p* < 0.001 in Tukey’s test) but were unaffected in the *rad4B* DM compared to the control strains (*p* > 0.05 in Tukey’s test).

Intriguingly, 40 h of dark incubation of the control strains’ conidia exposed to the UVB doses of 0.2, 0.3 and 0.4 J/cm^2^ resulted in the respective germination rates of 100%, 59% (±3.2) and 22% (±3.7), which were similar to the corresponding estimates of the *rad4B* DM (Figure 4C). In contrast, the rate of the *rad4A* DM’s conidia reactivated by 40 h dark incubation was only 1.3% (±0.6) at 0.2 J/cm^2^, 0.4% (±0.6) at 0.3 J/cm^2^ and zero at 0.4 J/cm^2^. These data indicate a high activity of Rad4A in the photoreactivation of conidia severely impaired or inactivated by UVB, and an infeasibility of its NER activity in the reactivation of those conidia in the field where daily nighttime is too short for NER.

Photoreactivation depends on the rapid photorepair of CPD and 6-4PP DNA lesions by Phr1 and Phr2 and their regulators WC1 and WC2 in *B. bassiana* and *M. robertsii* [19,34,36]. The Rad4A-Rad23 and Rad23-WC2 interactions detected in this study and the Rad23-Phr2 interaction detected previously [34] reveal a link of Rad4A to Phr2 and WC2 via Rad23, which is nucleocytoplasmic shuttling under external stresses including UVB irradiation [34]. The disruption of *rad4A* implicated a blockage of the link by inhibited protein–protein interactions and an impact on the expression levels of those genes required for photorepair. This implication was clarified by their expression levels. As illustrated in Figure 4D, *phr2* and *wc1* were downregulated by at least onefold in the *rad4A* DM versus WT cultures, accompanied by *wc2* and *rad23* being less downregulated. None of the analyzed genes were significantly upregulated. The downregulated genes shed light on the marked reduction in the *rad4A* DM’s photoreactivation activity.

## 4. Discussion

As presented above, both ad4A and Rad4B were nucleus-specific proteins proven to interact with the nucleocytoplasmic shuttling Rad23 [34] and played a little role in the in vitro and in vivo lifecycle of *B. bassiana*. The uniquely anti-UV role of Rad4A proved to be dependent on a capability of its reactivating UVB-impaired or UVB-inactivated conidia under visible light rather than in the dark within daily nighttime hours. Rad4A did show NER activity, which was similar to its homolog in the model yeast [21,44], but was insufficient to reactivate conidia lightly impaired at 0.1 J/cm^2^ by the end of 12 or 24 h dark incubation and was incapable of reactivating the conidia impaired at ≥0.2 J/cm^2^ even by the end of 40 h dark incubation. This highlights the infeasible NER activity of Rad4A for *B. bassiana* to resist solar UV under field conditions. In contrast, Rad4B was shown to have neither photoreactivation nor NER activity in this study, suggesting its completely functional redundancy in *B. bassiana* and no need for further discussion on it.

The upper limit of *B. bassiana*’s tolerance to UVB damage is ~0.5 J/cm^2^, which is roughly 5-fold lower than a solar UVB dose accumulated from the morning to 3:00 pm on a sunny summer day and is exploited to the proposal of a low-risk or non-risk strategy for the field application of a fungal pesticide between 3:00 pm and 5:00 pm or after 5:00 pm [8]. The proposed strategy is based on a neglected solar UVB dose accumulated after 5:00 pm and such a damage of ~0.2 J/cm^2^ accumulated between 3:00 pm and 5:00 pm that can be readily photoreactivated by visible light prior to the evening. In *B. bassiana*, Rad23 was previously shown to have not only a pleiotropic effect on the fungal lifecycle in vitro and in vivo but an essential anti-UV role since its null mutant suffered a decrease of 84% in conidial UVB resistance (LD_50_) and of 95% in photoreactivation activity, which was revealed by a 4.5% versus 82% photoreactivation rate in the null mutant versus the WT strain [34]. The present *rad4A* DM showed a similar loss of conidial UVB resistance (LD_50_ decreased by 81–83%) but a much smaller reduction in photoreactivation activity. The previous and present studies have demonstrated that Rad4A plays as important a role as Rad23 in sustaining conidial UVB resistance but was less efficient than Rad23 in photoreactivating the UVB-impaired conidia of *B. bassiana* deficient in the Rad23 homolog. This scenario is distinct from the Rad4-Rad23-Rad33 complex, in which only Rad4 is essential to the yeast cells surviving UV by means of NER while the roles of Rad23 and Rad33 are limited to preventing Rad4 from degradation and binding to Rad4 [30,31,32,33,44,45], respectively. This marked difference could be ascribed to the fact that Rad4A has evolutionarily been linked to photorepair-required Phr2 and WC2 through a bridge of its interaction with Rad23 and has acquired much stronger activity of photoreactivation than of its original anti-UV role depending on NER, which has devolved to a status unable to protect *B. bassiana* from solar UV damage in the field.

Aside from Rad4A and Rad23, anti-UV proteins orthologous to the yeast Rad1 and Rad10 have also proved to be essential for photoreactivation but infeasible for reactivation of conidia severely impaired by UVB through NER in *B. bassiana* and *M. robertsii* [36,38]. In previous studies, the null mutants of *rad1* and *rad10* displayed conidial UVB resistance reduced by 79% and 80% in *B. bassiana* and 92% and 94% in *M. robertsii*; photoreactivation rates of their conidia impaired at the UVB dose of 0.4 J/cm^2^ dropped to 3.3% and 4.3% from 87% of the *B. bassiana*’s WT strain and to an undetectable level from 63% of the *M. robertsii*’s WT strain. The extraordinarily high activities of Rad1 and Rad10 were considered to arise from the interactions of either Rad1 or Rad10 with both WC1 and WC2 serving as regulators of two photolyases in *B. bassiana* [36] and the Rad1-WC2 and Rad10-Phr1 interactions in *M. roberstii* [38]. The present study revealed an interaction only between Rad4A and Rad23 previously shown to interact with Phr2 [34], suggesting weaker links of Rad4A than of Rad1 or Rad10 to the photolyases and the photolyase regulators. Perhaps for this reason, the null mutant of *rad4A* was much less compromised in photoreactivation activity than were the previous null mutants of *rad1* and *rad10*. Altogether, the present and previous studies provide strong evidence for the hypothesis that molecular mechanisms underlying filamentous fungal adaptation to solar UV aredistinct from those depending on NER for the model yeast cells’ survival of UV [26]. Therefore, the anti-UV proteins presumably functioning in the NER pathway to different degrees have evolutionarily acquired photoreactivation activity to protect filamentous fungal cells from solar UV damage.

Conclusively, our study on two Rad4 paralogs unveils that only Rad4A plays a vital role in the photoprotection of *B. bassiana* from solar UVB damage but no other role in the in vitro and in vivo fungal lifecycle. The vital role relies on the Rad4A-Rad23 interaction that links Rad4A to the photolyase Phr2 and the photolyase regulator WC2, which interacts with WC1, the other regulator of Phr1 and Phr2 [35,36]. Rad4B is similar to Rad4A in its nucleus-specific localization and interaction with Rad23 but has no anti-UV role at all in *B. bassiana*, implicating a status of its evolutionary remnant. This finding expands upon a molecular basis supporting filamentous fungal adaptation to solar UV irradiation on the Earth’s surface.

## Figures and Tables

**Figure 1 jof-09-00154-f001:**
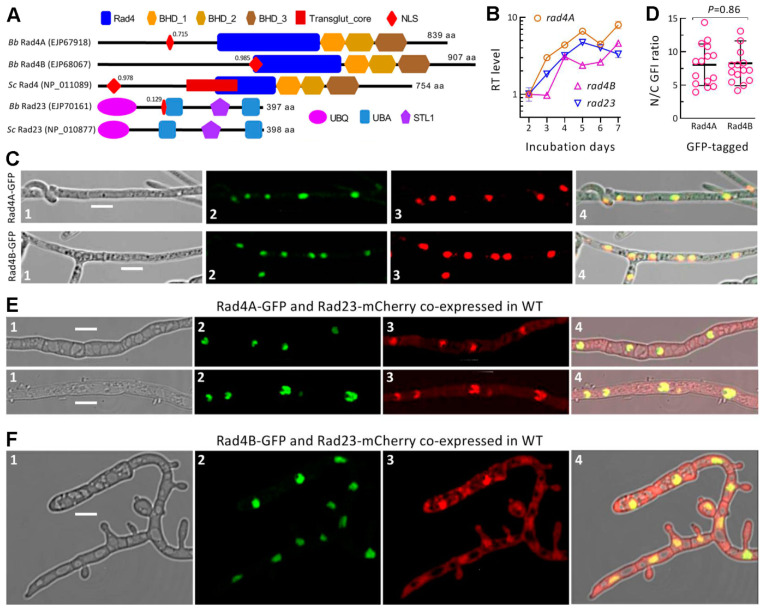
Recognition, expression and localization of Rad4A, Rad4B and Rad23 in *B. bassiana*. (**A**) Comparative domain architectures of Rad4 and Rad23 homologs in *B. bassiana* (*Bb*) and *S. cerevisiae* (*Sc*). Decimal value associated with the NLS motif of each protein is a maximal probability predicted for its nuclear localization. (**B**) Relative transcript (RT) levels of *rad4A*, *rad4B* and *rad23* in a wild-type (WT) *Bb* strain over the days of incubation (relative to Day 2) on SDAY at the optimal regime of 25 °C and 12:12 (L:D). (**C**) LSCM images (scale: 5 μm) for subcellular localization of Rad4A-GFP and Rad4B-GFP fusion proteins expressed in the WT hyphae at the optimal regime and stained with the nuclear dye DAPI (shown in red). Images 1–4 are bright, expressed, stained and merged views of the same microscopic field, respectively. (**D**) N/C-GFI ratios of Rad4A-GFP and Rad4B-GFP (*p*, Student’s *t* test). (**E**,**F**) LSCM images (scale: 5 μm) for nuclear localization of Rad4A-GFP and Rad4B-GFP (Image 2) co-expressed with Rad23-mCherry (Image 3) in the WT strain, respectively. Error bars: standard deviations (SDs) from three cDNA samples analyzed (**B**) or 15 hyphal cells (**D**).

**Figure 2 jof-09-00154-f002:**
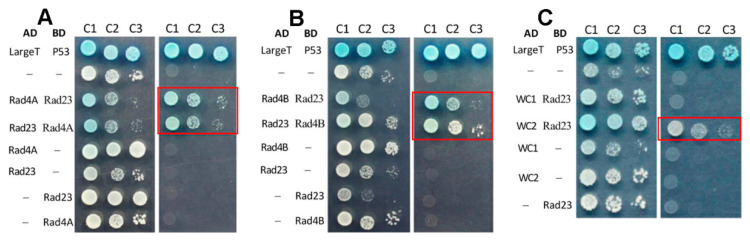
Y2H assays for an interaction of either Rad4A (**A**) or Rad4B (**B**) with Rad23 and of Rad23 with WC2 (**C**). Note that the target diploids (framed in red) and the positive control (AD-LargeT- BD-P53) grew well on the quadruple-dropout plate during a 3-day incubation at 30 °C after inoculation with 5 × 10^4^ (C1), 5 × 10^3^ (C2) and 5 × 10^2^ (C3) cells.

**Figure 3 jof-09-00154-f003:**
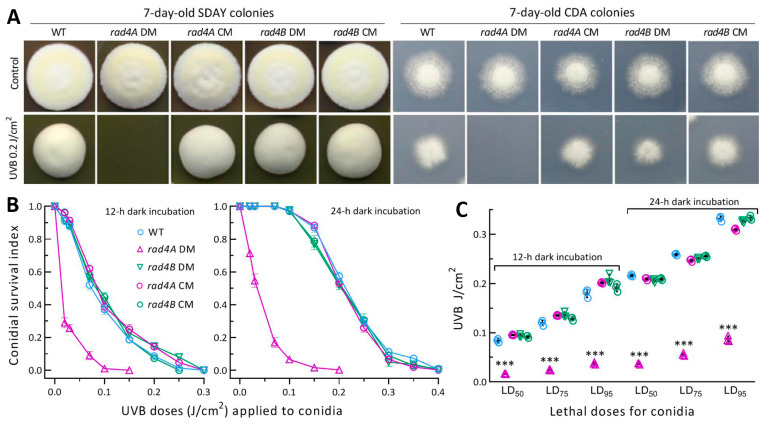
Essentiality of Rad4A versus dispensability of Rad4B for *B. bassiana*’s resistance to UVB irradiation. (**A**) Images for 7-day-old colonies of disruption mutants (DMs) and control (CM and WT) strains grown in the optimal regime after SDAY and CDA plates inoculated with ~10^3^ conidia were irradiated at the UVB dose of 0.2 J/cm^2^. (**B**) Survival trends of conidia incubated for 12 and 24 h in the dark after exposure to gradient UVB doses. (**C**) LD_50_, LD_75_ and LD_95_ estimated as indices of conidial UVB resistance from fitted survival trends. *** *p* < 0.0001 in Tukey’s HSD test. Error bars: SDs from three independent replicates.

**Figure 4 jof-09-00154-f004:**
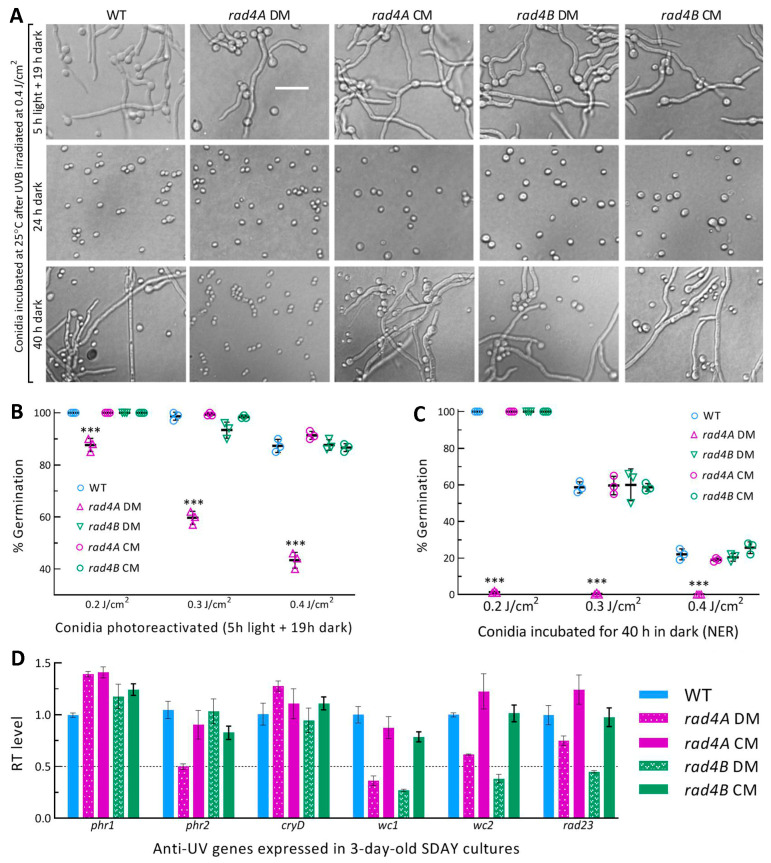
Essential role of Rad4A versus null role of Rad4B in photoreactivation of UBV-impaired conidia in *B. bassiana*. (**A**) Microscopic images (scale: 20 μm) for conidial germination status of disruption mutants (DMs) and control (CM and WT) strains incubated at 25 °C for 5 h under white light plus 19 h in the dark (photoreactivation) or directly for 24 and 40 h in the dark (NER) after exposure to the UVB dose of 0.4 J/cm^2^. (**B**,**C**) Germination percentages of the tested strains’ conidia photoreactivated at 25 °C for 5 h or reactivated by 40 h dark incubation (NER) at 25 °C after exposure to the indicated UVB doses. *** *p* < 0.0001 in Tukey’s HSD test. (**D**) Relative transcript (RT) levels of anti-UV genes in the 3-day-old SDAY cultures of *rad4A* and *rad4B* mutants with respect to the WT standard. The dashed line denotes a significant level of onefold downregulation. Error bars: SDs from three independent replicates.

## Data Availability

All experimental data are included in this paper and the Appendix A.

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
