# Peer review of "Comparative Roles of Rad4A and Rad4B in Photoprotection of Beauveria bassiana from Solar Ultraviolet Damage"

_jof, 2023, doi:10.3390/jof9020154_

Round 1

Reviewer 1 Report

I only have a few problems to be solved.

Please specify data analyses, One way analyses is ANOVA?

So the data were normally distributed? Why Student t test was used, this is a basic statistical analyses, It would be better an r.

Author Response

Comments and Suggestions for Authors

I only have a few problems to be solved.

Author response: Thank you very much reviewing and understanding our manuscript.

Please specify data analyses, One way analyses is ANOVA?

So the data were normally distributed? Why Student t test was used, this is a basic statistical analyses, It would be better an r.

Author response: Analysis of variance (ANOVA) is usually applied in one or two ways to reveal variation(s) of one or two factors. In our study, one-way ANOVA was performed to assess a variation of experimental data among tested null mutants and control strains. When the tested factor contains only two treatments, a pair of data sets can be analyzed using the Student's t test rather than the one-way ANOVA required for at least three treatments (fungal strains in our study). The mentioned r should be a correlation coefficient between two factors and means nothing for our situation (comparison of two fusion proteins expressed in the same WT strain (factor)).

Reviewer 2 Report

Manuscript ID: jof-2156704

Title: Comparative Roles of Rad4A and Rad4B in Photoprotection of Beauveria bassiana from Solar Ultraviolet Damage

Dear Author

Reviewer comments:

 About Beauveria bassiana, it has an effective role in the biological control of many diseases. In this research, components produced from this fungus were analyzed as secondary products that have an effect on pathogens, which would reduce the number of harmful pathogens. This paper contains interesting studies on characterizing The Rad4-Rad23-Rad33 complex plays an essential anti-ultraviolet (UV) role depending on nucleotide excision repair (NER). Although many studies have been conducted on that is still lacking. In addition, this study is significant for the experimental regions. However, this manuscript still needs improving in writing logic, and some language expressions and the analysis of the discussion are not clear. It is recommended to modify the language for a better version. Moreover, the discussion section is not deep enough.

Abstract:
-it is good, but it short, the authors should consider the proposed changes for improving the clarity of the content. Such add the background on Beauveria bassiana and its effectsof NER activity of Rad4A in the field

Keyword: good

-Introduction

part is appropriate but a few things are needed for further improvements especially the study aims should be added. Update the new literature. Upon field application. Upon field application, formulated conidia are exposed to solar ultra-formulated conidia are exposed to solar ultra violet (UV) irradiation composed of UVA and UVB wavelengths. Add some studies about the study with highlighting research gaps, which necessitated conducting this trial. And we need know the mode of action in DNA repair of UV irradiation and Rad4-Rad23-Rad33 proteins

Materials and methods:
-this part describes very well by using suitable subheadings. However, it needs few modifications and details of selecting primers and amplification conditions in the revised version to enhance

Need details about isolation of Fungal Rad4 Homologs? If you are doing them.

Results and Discussion
- This part is clearly presented

-Both parts need to combine if possible or as Journal style and it needs minor revision and it needs to discuss the effects of UV irradiation on the fungus under heat stress it may be added in RT-PCR experiments  

Figures S1 and 2 need to add in results as Figures to be clearer for readers

Conclusion: good  

References:
-Cross-check the references in the text and reference cite. Few references are not as per journal style in the text as well reference section

Author Response

Reviewer comments:

 About Beauveria bassiana, it has an effective role in the biological control of many diseases. In this research, components produced from this fungus were analyzed as secondary products that have an effect on pathogens, which would reduce the number of harmful pathogens. This paper contains interesting studies on characterizing The Rad4-Rad23-Rad33 complex plays an essential anti-ultraviolet (UV) role depending on nucleotide excision repair (NER). Although many studies have been conducted on that is still lacking. In addition, this study is significant for the experimental regions. However, this manuscript still needs improving in writing logic, and some language expressions and the analysis of the discussion are not clear. It is recommended to modify the language for a better version. Moreover, the discussion section is not deep enough.

Author response: Thank you very much for reviewing and understanding our manuscript. Since documented information on filamentous fungal NER activity still is very limited, the discussion cannot go too far based on what we have learned so far. Our present and previous studies highlight that the NER activities of anti-UV RAD proteins characterized in model yeast are not feasible in B. bassiana as a filamentous fungal insect pathogen.

Abstract:
-it is good, but it short, the authors should consider the proposed changes for improving the clarity of the content. Such add the background on Beauveria bassiana and its effects of NER activity of Rad4A in the field

Author response: The abstract has been rewritten for clarity.

Keyword: good

-Introduction

part is appropriate but a few things are needed for further improvements especially the study aims should be added. Update the new literature. Upon field application. Upon field application, formulated conidia are exposed to solar ultra-formulated conidia are exposed to solar ultra violet (UV) irradiation composed of UVA and UVB wavelengths. Add some studies about the study with highlighting research gaps, which necessitated conducting this trial. And we need know the mode of action in DNA repair of UV irradiation and Rad4-Rad23-Rad33 proteins

Author response: The Introduction has compiled sufficient information on the backgrounds of photorepair and NER of UV-induced DNA lesions in filamentous and yeast fungi and also pointed out a big gap of knowledge about NER that has been rarely explored in fungi other than the budding yeast. The cited references in the mentioned sentence are pertinent.

Materials and methods:
-this part describes very well by using suitable subheadings. However, it needs few modifications and details of selecting primers and amplification conditions in the revised version to enhance

Author response: For conciseness, we did not try to mention the names of many primers we used in the study. Instead, we provided detailed information on each pair of primers in Tables S1 and S2, including the purpose of its use.

Need details about isolation of Fungal Rad4 Homologs? If you are doing them.

Author response: The amplification of a target gene from the genomic DNA or cDNA of a fungal strain through PCR is just a routine with no need of detail as long as its accession code and used primers are given.

Results and Discussion
- This part is clearly presented

-Both parts need to combine if possible or as Journal style and it needs minor revision and it needs to discuss the effects of UV irradiation on the fungus under heat stress it may be added in RT-PCR experiments

Author response: No need of combination. The journal style demands a separation of the Results section from the Discussion section. Fungal responses to UV and heat shock are controlled by different pathways and hardly discussed together although the two stresses are often concurrent in summer. The effect of UV irradiation the fungal response to heat stress is out of the purpose of our study because the studied genes are putatively anti-UV genes and not involved in heat tolerance.

Figures S1 and 2 need to add in results as Figures to be clearer for readers

Author response: Keeping Figures S1 and S2 in Supplementary Material has no impact on the accession by interesting readers.

Conclusion: good  

References:
-Cross-check the references in the text and reference cite. Few references are not as per journal style in the text as well reference section

Author response: We have double checked all references and citations following the journal style.